# 24,25-Dihydroxy Vitamin D and Vitamin D Metabolite Ratio as Biomarkers of Vitamin D in Chronic Kidney Disease

**DOI:** 10.3390/nu15030578

**Published:** 2023-01-22

**Authors:** Seunghye Lee, Hye Jin Chung, Sehyun Jung, Ha Nee Jang, Se-Ho Chang, Hyun-Jung Kim, Min-Chul Cho

**Affiliations:** 1Division of Nephrology, Department of Internal Medicine, Gyeongsang National University Hospital, Gyeongsang National University College of Medicine, Jinju 52727, Republic of Korea; 2College of Pharmacy and Research Institute of Pharmaceutical Sciences, Gyeongsang National University, Jinju 52727, Republic of Korea; 3Institute of Health Sciences, Gyeongsang National University, Jinju 52727, Republic of Korea; 4Department of Laboratory Medicine, Gyeongsang National University Hospital, Gyeongsang National University College of Medicine, Jinju 52727, Republic of Korea

**Keywords:** 24,25-dihydroxy vitamin D, 25-hydroxyvitamin D, vitamin D metabolite ratio, chronic kidney disease

## Abstract

The appropriate management of vitamin D deficiency and hyperparathyroidism is essential to prevent metabolic bone disorder (MBD) and cardiovascular diseases in chronic kidney disease (CKD). Recently, the 24,25-dihydroxyvitamin D [24,25(OH)_2_D] and vitamin D metabolite ratio (VMR), i.e., the ratio of 24,25(OH)_2_D to 25-hydroxyvitamin D [25(OH)D], have emerged as biomarkers of vitamin D level. We analyzed the usefulness of vitamin D biomarkers for the evaluation of MBD in patients with CKD. We analyzed blood and urine samples from 208 outpatients with CKD stage G2–G5. 25(OH)D showed a poor correlation with the estimated glomerular filtration rate (eGFR). Conversely, the 24,25(OH)_2_D level and VMR were significantly correlated with eGFR and the intact parathyroid hormone level. In conclusion, 24,25(OH)_2_D and VMR have the potential to be vitamin D biomarkers for the detection of MBD in CKD patients.

## 1. Introduction

Vitamin D maintains bone health by regulating serum calcium and phosphate levels. It also regulates the immune system and inhibits cancer, angiogenesis, and inflammatory responses. Vitamin D deficiency is associated with the development of various diseases, such as autoimmune diseases, cancers, cardiovascular diseases, and diabetes [1,2]. Vitamin D is obtained through sunlight exposure of the skin; it is converted into an active form through two-step hydroxylation in the liver and kidneys [3]. An activated form of vitamin D, 1,25 dihydroxy vitamin D [1,25(OH)_2_D], is produced by 1α-hydroxylase in the kidney [4]. Impaired vitamin D, calcium, and phosphorus metabolism occurs early in chronic kidney disease (CKD). One of the underlying mechanisms is increased parathyroid hormone (PTH) secretion [5]. The intestinal absorption of calcium, and the renal excretion of calcium and phosphate are important actions of 1,25(OH)_2_D. With CKD progression, the level of 1,25(OH)_2_D is decreased, leading to hypocalcemia and hyperphosphatemia, which further increases PTH secretion and exacerbates secondary hyperparathyroidism (SHPT). Therefore, a normal vitamin D level is essential to prevent and treat SHPT [6].

Reduced activity of 1α-hydroxylase leads to chronic vitamin D deficiency in CKD, causing SHPT and deranged serum calcium and phosphorus levels [7]. SHPT causes bone disease and consequently increases the risk of fractures. Furthermore, it leads to vascular calcification due to high serum calcium and phosphorus levels, thereby increasing the mortality rate of CKD patients [8].

The accurate assessment of the vitamin D level is essential for the prevention and treatment of SHPT through the correction of electrolyte imbalance and vitamin D supplementation. Serum total 25(OH)D level is commonly used to determine the vitamin D status. However, there have been debates about the reliability of this biomarker. It consists of the vitamin D binding protein (VDBP)-bound 25(OH)D, albumin-bound 25(OH)D, and free 25(OH)D, and they may be influenced by ethnicity, religion, sunlight exposure, and several conditions altering vitamin D metabolism, such as obesity, drugs, and diabetes, in addition to CKD [9,10]. In addition, previous observational studies and randomized trials have suggested that serum 25(OH)D may not be an accurate biomarker of vitamin D status, especially in bone health [11,12,13]. In fact, the 25-hydroxylation reaction in the liver does not involve a delicate feedback process to control the vitamin D level in the body, whereas the process of 1α-hydroxylation synthesizing 1,25(OH)_2_D in the kidney involves many feedback processes via the vitamin D receptor (VDR) [4]. In this respect, the biomarker that most accurately reflects vitamin D status in the body would be 1,25(OH)_2_D. 1,25(OH)_2_D is known to play an important role in vitamin D metabolism and functionally interacts with PTH or fibroblast growth factor-23 (FGF-23) in CKD. However, it has a very short half-life of about 5 to 8 h; thus, its concentration varies greatly during the day. In addition, since the blood concentration of 1,25(OH)_2_D is relatively low, it is difficult to accurately measure its serum concentration [14,15]. Thus, the KDIGO guideline of CKD-MBD does not recommend the routine measurement of 1,25(OH)D_2_D in CKD patients [16,17]. 

Recently, serum 24,25(OH)_2_D, and the ratio of 24,25(OH)_2_D and 25(OH)D levels (vitamin D metabolite ratio [VMR]) have been proposed as potential alternative biomarkers of the body’s vitamin D level [18,19]. 24,25(OH)_2_D is a catabolic product of a negative feedback mechanism to control the concentration of vitamin D in the blood, and this mechanism is complicated. In case of vitamin D toxicity, negative feedback turns on the catabolism mechanism to reduce blood vitamin D levels. The activated 24(OH)D hydroxylase enzyme (CYP24A1) converts 25(OH)D to 24,25(OH)_2_D by the stimulation of 1,25(OH)_2_D [20,21]. Therefore, 24,25(OH)_2_D or VMR could be a physiologically better biomarker than the currently used 25(OH)D, in that they represent the result of a feedback mechanism. In addition, previous studies have suggested that VMR may have a stronger association with bone health than that of 25(OH)D [22,23]. The kidney is a major organ in which CYP24A1 are expressed and activated. When evaluating vitamin D status in CKD patients, it may be necessary to consider not only 25(OH)D level, but also that the activity of CYP24A1 may change depending on the kidney function. For this reason, 24,25(OH)_2_D level or VMR might be biomarkers of vitamin D status, particularly in CKD patients [24].

There are rare studies comparing vitamin D biomarkers, including 25(OH)D, 24,25(OH)_2_D, and VMR in patients with CKD. In this present study, we compared various vitamin D biomarkers according to renal function in CKD patients, as well as the association of the biomarkers with intact PTH (iPTH). Since there are several limitations to the current method for evaluating vitamin D status with 25(OH)D, we aim to evaluate whether 24,25(OH)_2_D or VMR can be a biomarker alternative to 25(OH)D in CKD.

## 2. Materials and Methods

### 2.1. Study Participants

This prospective study enrolled outpatients with CKD aged >18 years who visited the Department of Nephrology, Gyeongsang National University Hospital, between June and December 2021. We analyzed a subset of blood samples from a total of 208 patients who provided informed consent. Patients taking calcimimetics, phosphate binders, any medications and supplements containing calcium, or phosphorus, as well as vitamin D, were excluded. This study was approved by the Institutional Review Board of Gyeongsang National University Hospital (approval no.: GNUH 2021-04-015).

### 2.2. Vitamin D Level Measurement and Calculation of VMR

The serum concentration of 25(OH)D and 24,25(OH)_2_D was measured by the method mentioned in a previous study with slight modifications [25]. For the measurement, liquid chromatography with mass spectrometry (LC-MS/MS) was used after the solid-phase extraction (SPE) method. 24,25(OH)_2_D can currently be measured only by the LC-MS/MS method. While 25(OH)D can also be measured by a electrochemiluminescence binding assay, it was measured by the LC-MS/MS method for consistency in the measurement method. Following the addition of the internal standard, stable isotope-labeled d6-24,25(OH)_2_D and d6-25(OH)D were added to 200 μL of serum sample. After this, methanol was mixed with the solution. To precipitate protein, the solution was mixed by vortexing and kept at 4 °C for 10 min. After centrifugation at 4 °C and 12,000 *g* for 10 min, phosphate-buffered saline was added into the supernatant and applied onto an SPE cartridge. After performing SPE, the vacuum system was applied to the eluted fraction for evaporation. The dried remnant was reconstituted in 75% methanol. In total, 5 μL of the reconstituted solution was analyzed by the LC-MS/MS system. The LC-MS/MS system in our study was an Agilent 1260 HPLC system (Agilent Technologies, Palo Alto, CA, USA) with an Agilent 6460 triple quadrupole mass spectrometer (Agilent Technologies) and an electrospray ionization source. For the separation of 24,25(OH)_2_D and 25(OH)D by HPLC, a Kinetex^®^ Biphenyl column (2.6 μm, 3.0 × 100 mm; Phenomenex, Torrance, CA, USA) and Poroshell^®^ 120 EC-C 18 column (2.7 μm, 3.0 × 50 mm; Agilent Technologies) were utilized, respectively. The mobile phase in HPLC used 0.1% aqueous formic acid and methanol, and was processed by a gradient program at a flow rate of 0.4 mL/min. The monitored transitions were 417 → 381 *m*/*z* for 24,25(OH)_2_D, 423 → 387 *m*/*z* for d6-24,25(OH)_2_D, 401 → 383 *m*/*z* for 25(OH)D, and 407 → 389 *m*/*z* for d6-25(OH)D. In the LC-MS/MS system for the present study, the limits of quantitation of 24,25(OH)_2_D and 25(OH)D were 0.2 and 2 ng/mL, respectively.

The VMR was calculated according to a previous study [26]. It was calculated by dividing 24,25(OH)_2_D concentration by 25(OH)D concentration and then multiplying it by 100.

### 2.3. Demographics and Laboratory Findings

The following data were collected through the Electrical Medical Record (EMR) to analyze the correlation between the CKD stage and vitamin D biomarkers in the patients with CKD. Age, sex, height, serum creatinine (SCr), serum albumin, urine protein to creatinine ratio (uPCR, mg/g), serum calcium (mg/dL), serum phosphate (mg/dL), calcium–phosphate product (Ca × P, mg^2^/dL^2^), and iPTH were recorded. Medical records were screened to identify patients with a history of diabetes mellitus and hypertension. The estimated glomerular filtration rate (eGFR) was calculated using the 2012 CKD-EPI creatinine equation. The CKD stage was classified as G2-G5, according to the Kidney Disease: Improving Global Outcomes (KDIGO) Clinical Practice Guidelines [27,28]. The cause of CKD was classified as glomerular disease (diabetic nephropathy and chronic glomerulonephritis), tubulointerstitial nephritis, vascular disease, polycystic kidney disease, and others. Hyperparathyroidism was defined as iPTH level >65 pg/mL.

### 2.4. Statistical Analysis

To assess the correlation between two variables, Pearson’s correlation test was used. For variables with a normal distribution, differences between the groups were evaluated using an analysis of variance. For multivariate analysis, multivariate linear regression and logistic regression analyses were used. We confirmed the goodness-of-fit of the regression models using an adjusted R-squared (R^2^), the Akaike information criterion, and the Bayesian information criterion. All the tests were two tailed, and *p* < 0.05 was considered statistically significant. We used R software (version 4.0.3; R Foundation for Statistical Computing, Vienna, Austria) and SPSS software (version 21.0; IBM Corp., Armonk, NY, USA) for statistical analyses and the construction of graphics.

## 3. Results

### 3.1. Demographics and Laboratory Findings of Study Participants

Table 1 summarizes the demographics and laboratory findings of the participants of this study. The mean age of the 208 patients was 64.3 ± 12.7 years, and 133 (63.9%) patients were males. Diabetes mellitus and hypertension were present in 167 (80.3%) and 98 (47.1%) patients, respectively. The most common cause of CKD was chronic glomerulonephritis (33.8%), followed by chronic tubulointerstitial nephritis (27.4%), diabetic nephropathy (21.6%), vascular disease (17.8%), and polycystic kidney disease (3.8%). The mean eGFR and serum creatinine level were 39.38 ± 17.47 mL/min/1.73 m^2^ and 2.00 ± 1.07 mg/dL, respectively. The eGFR was >60 mL/min/1.73 m^2^ in 13.9% of patients. The mean serum Ca, P, and iPTH levels were 9.38 ± 0.58, 3.76 ± 0.78, and 58.49 ± 44.41 pg/mL, respectively. The numbers of patients with SHPT in stage G2, G3a, G3b, G4, and G5 were 6 (20.7%), 5 (10.2%), 18 (29.0%), 19 (35.2%), and 8 (57.1%), respectively.

### 3.2. iPTH and Vitamin D Biomarkers According to CKD Stage

Table 2 shows the changes in iPTH, 24,25(OH)2D, 25(OH)D, and VMR levels, according to the CKD stage. The mean values of 25(OH)D, 24,25(OH)_2_D, and VMR levels were 22.81 ± 11.24 ng/mL, 0.83 ± 0.63 ng/mL, and 3.55 ± 1.91, respectively. In CKD stage G3a, the mean iPTH level was 42.10 ± 20.43 pg/mL, which was lower than that of G2. However, in general, the mean iPTH level increased as the CKD stage increased from G2 to G5 (*p* < 0.001). The 25(OH)D level did not differ according to the CKD stage (*p* = 0.353), whereas the 24,25(OH)2D level and VMR showed a significant difference according to the CKD stage (*p* = 0.011 and <0.001, respectively).

### 3.3. Correlations of eGFR with iPTH Level and Vitamin D Biomarkers

A negative correlation was found between iPTH level and eGFR (R^2^ = 0.0874, *p* < 0.001) (Figure 1a). The 25(OH)D level showed no significant correlation with the eGFR (R^2^ = 0.0011, *p* = 0.632) (Figure 1b). There was a positive correlation between the eGFR and the 24,25(OH)_2_D level (R^2^ = 0.0603, *p* < 0.001). The eGFR and VMR were also positively correlated (R^2^ = 0.1523, *p* < 0.001) (Figure 1c,d).

### 3.4. Correlations of iPTH Level with Vitamin D Biomarkers

The iPTH level was negatively correlated with the 25(OH)D (R^2^ = 0.0277, *p* = 0.016) (Figure 2a), 24,25(OH)_2_D (R^2^ = 0.0718, *p* < 0.001) (Figure 2b), and VMR (R^2^ = 0.0689, *p* < 0.001) (Figure 2c) levels. The serum phosphate level was negatively correlated with the eGFR. The serum calcium, phosphate, and Ca × P levels were not correlated with the iPTH level or vitamin D biomarkers (Table A1). 

### 3.5. Vitamin D Biomarkers According to eGFR and iPTH Level

Figure 3 shows the changes in the iPTH, 25(OH)D, 24,25(OH)_2_D, and VMR levels, according to the CKD stage. The mean iPTH level was the lowest in CKD stage G3a and gradually increased with the increasing stage to CKD stage G5, except in CKD stage G2. However, the 24,25(OH)_2_D level and the VMR decreased with the increasing CKD stage, from CKD stage G2; this trend was more prominent for the VMR. The 25(OH)D level did not change with the 24,25(OH)_2_D level or the VMR between different CKD stages.

Nine multiple regression analysis models were created to explain the relationships between the three vitamin D biomarkers and the eGFR or iPTH. The fit of each model was compared using the adjusted R^2^, the Akaike information criterion, and the Bayesian information criterion. In models using the eGFR as a dependent variable (models 4–6), model 6, which explained the eGFR using the VMR, had the best fit. Of the three vitamin D biomarkers, the VMR was the most strongly associated with the eGFR. In the models using iPTH as a dependent variable (models 7–9), model 8, which explained iPTH using 24,25(OH)_2_D, had the best fit. Of the three vitamin D biomarkers, 24,25(OH)_2_D was most strongly associated with the iPTH level (Table 3). 

### 3.6. Vitamin D Biomarkers and Bone Density

In total, 43 of 208 patients performed bone densitometry. The result is provided in Table 4. 

The eGFR, 25(OH)D, 24,25(OH)_2_D and VMR did not show a significant difference according to bone density. In osteoporosis patients, the level of vitamin D biomarkers tended to decrease. 

## 4. Discussion

In the present study, we compared biomarkers of the vitamin D level in patients with CKD, particularly to determine the usefulness of 24,25(OH)_2_D and VMR as the alternative biomarkers of 25(OH)D. 

As shown in Table 2, the mean iPTH level of patients with CKD stage G2 was 52.12 pg/mL, which was higher than that of stage G3a (42.10 pg/mL). The 25(OH)D level was higher in stage G2 than in stage G3a. However, the 24,25(OH)_2_D level and VMR tend to decrease consistently from stage G2 to G5. Although various biochemical indicators related to CKD-MBD are altered in CKD stage 3, clinical features, such as degree of severity and rate of change, vary among patients. The KDIGO guideline recommends that serum calcium, phosphorus, and PTH levels should be monitored from CKD stage 3 in adults and from CKD stage 2 in children [17].

In children with slowly progressive kidney disease, the PTH level may be elevated from CKD stage 2 [29,30]. In a previous study, SHPT was diagnosed in 12% of CKD stage G2 patients, with an eGFR of 80 mL/min/1.73 m^2^; additionally, the proportion of patients with SHPT increased rapidly with a decreasing eGFR [31]. Therefore, metabolic changes associated with SHPT, including low vitamin D level, may occur during the early stage of CKD. As a result, we included CKD stage 2 patients in the present study. Our results showed that the 24,25(OH)_2_D level and VMR were more strongly correlated with eGFR than iPTH or the 25(OH)D level, particularly in the early stages of CKD.

Levin et al. [31] found lower 25(OH)D levels in more advanced CKD stages. The KDIGO CKD-MBD guideline suggests that the 25(OH)D level is the preferred indicator of the body stores of vitamin D in CKD patients [17]. However, in Figure 2 of this presented study, the 24,25(OH)_2_D level and the VMR, but not the 25(OH)D level, were significantly correlated with the KDIGO CKD stage and the eGFR; this suggests that the 24,25(OH)_2_D level and the VMR are better indicators of vitamin D level than 25(OH)D, the currently used vitamin D biomarker in CKD patients. In several previous studies, the 24,25(OH)_2_D, VMR and 25(OH)D have been controversial as vitamin D biomarkers. A study found that VMR may be used as a complementary indicator of vitamin D level for the evaluation of endogenous metabolism in premenopausal women [32]. In a study of vitamin D biomarkers conducted in the US, the VMR was similar in blacks and whites; conversely, the 25(OH)D level was negatively correlated with the iPTH level in both races [26]. On the contrary, the VMR does not show significant differences between races and is known to be a better biomarker than 25(OH)D, especially in cases of vitamin D deficiency [33]. Therefore, it is possible to apply the VMR to other ethnic groups.

Although all the evaluated vitamin D biomarkers, including the 25(OH)D, 24,25(OH)_2_D, and VMR, were significantly correlated with the iPTH level, the 24,25(OH)_2_D level showed the strongest correlation with the iPTH level in the model fitness analysis. A previous study showed that the 24,25(OH)_2_D level had a stronger association with iPTH than the 25(OH)D level [16]. However, there is still insufficient evidence to confirm whether 24,25(OH)_2_D and the VMR are better predictors of the effects of vitamin D supplementation on MBD than 25(OH)D, particularly in CKD patients. Our study suggests that 24,25(OH)_2_D and the VMR may be more sensitive vitamin D biomarkers than 25(OH)D in CKD patients, because they started to decrease at an earlier stage than 25(OH)D in Figure 3.

Some studies suggest that 24,25(OH)_2_D or the VMR is a better biomarker than 25(OH)D, especially for bone health. Ginsberg et al. showed that the VMR had stronger correlation with both BMD and fracture risk compared to 25(OH)D [11]. Another study found that the serum 25(OH)D level remained unchanged throughout the entire study period, despite taking vitamin D supplements, but that the bone density showed a dose-dependent effect on Vitamin D [12]. This suggested that 25(OH)D might not accurately represent vitamin D status and bone health. However, the reasons why the effect of 24,25(OH)_2_D or the VMR differs from 25(OH)D in bone health are unclear. In Table 4 of this presented study, we analyzed the results of bone densitometry in some of our study participants, and vitamin D biomarkers did not show a significant difference according to bone density. However, there is a limitation, in that the number of patients was too small. A follow-up study including more data about bone densitometry seems necessary to evaluate the correlation between bone density and vitamin D biomarkers. 

In general, the synthesis of 1,25(OH)_2_D is decreased in CKD due to a reduced 1α-hydroxylase function. The vitamin D level is regulated by a precise feedback mechanism that maintains homeostasis between its production and catabolism [34,35]. The CYP24A1 enzyme plays a major role in vitamin D catabolism, is induced by 1,25(OH)_2_D and FGF-23, and is inhibited by PTH [36,37,38]. During 25(OH) catabolism by CYP24A1, the most abundant product is 24,25(OH)_2_D [39]. Therefore, 24,25(OH)_2_D reflects the vitamin D level more accurately than 25(OH)D, particularly because it is an indicator of catabolism that maintains the concentration of active vitamin D in vivo through feedback. Additionally, 24,25(OH)_2_D is an appropriate vitamin D biomarker. First, the circulating half-life of 24,25(OH)_2_D in blood is relatively long (i.e., almost 7 days). Second, it has a relatively high level, which is why it is measured in ng/mL [39]. For comparison, 25(OH)D, the currently used vitamin D biomarker, also has a high blood level and is measured in ng/mL.

This study has several strengths. First, 24,25(OH)2D and the VMR were compared with 25(OH)D as vitamin D biomarkers in the clinical setting with CKD patients. Second, our study included patients of various CKD stages from G2 to G5, suggesting the possibility of 24,25(OH)2D and the VMR as a potential vitamin D biomarker in a wide range of CKD situations. Third, the measurement of 25(OH)D in a clinical laboratory commonly proceeds with an electrochemiluminescence assay, a type of immunoassay. However, in our study, the measurement of 25(OH)D and 24,25(OH)2D was performed with the same test platform, LC-MS/MS, to minimize errors due to differences in the analyzers and to increase the accuracy. 

Our study had two main limitations. First, this was a cross-sectional study, and follow-ups of patients or blood investigations were not performed. Second, several factors that could affect the serum vitamin D level, including seasonal variation, food, outdoor activity duration, and use of sunscreen, were not analyzed. However, to minimize seasonal variation, we conducted the study over two seasons, from summer to autumn, in South Korea. According to the website, which provides year-round weather statistics for Seoul, the capital of South Korea, there is an average of 2428 h of sunlight per year (of a possible 4383) with an average of 6:38 of sunlight per day [40]. Because of these limitations, it was not possible to eliminate all the confounding factors. Therefore, to further clarify our findings, more sophisticated and large-scale research might be required.

## 5. Conclusions

Although 25(OH)D is the most commonly performed test to assess vitamin D levels in CKD patients, it was poorly correlated with the eGFR. On the other hand, 24,25(OH)_2_D and the VMR were significantly correlated with the eGFR and iPTH levels in patients with CKD stage G2–G5. Therefore, we suggest that 24,25(OH)_2_D and the VMR have the potential to be alternative vitamin D biomarkers to 25(OH)D for the diagnosis of MBD in CKD patients.

## Figures and Tables

**Figure 1 nutrients-15-00578-f001:**
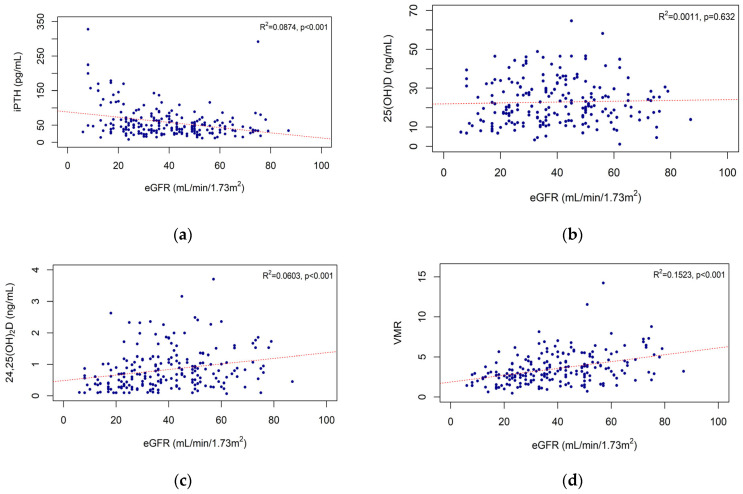
Correlations of eGFR with iPTH and vitamin D biomarkers: eGFR and iPTH (**a**), eGFR and 25(OH)D (**b**), eGFR and 24,25(OH)_2_D (**c**), and eGFR and VMR (**d**). Abbreviations: eGFR, estimated glomerular filtration rate; iPTH, intact parathyroid hormone; VMR, vitamin D metabolites ratio; 25(OH)D, 25-hydroxy vitamin D; 24,25(OH)_2_D, 24,25-dihydroxy vitamin D.

**Figure 2 nutrients-15-00578-f002:**
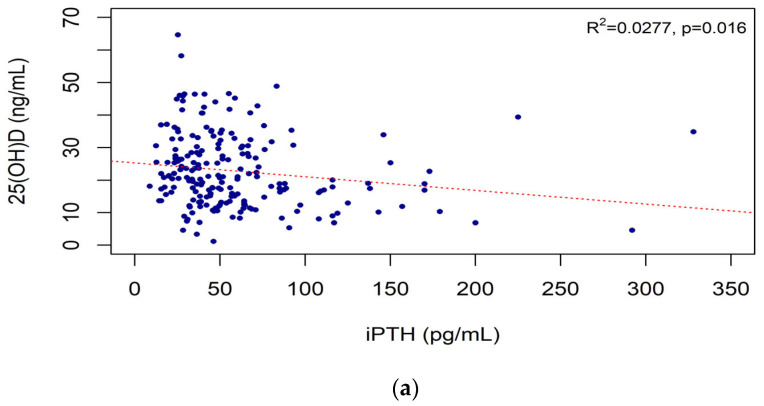
Correlations of intact PTH with vitamin D markers: iPTH and 25(OH)D (**a**), iPTH and 24,25(OH)2D (**b**), and iPTH and VMR (**c**). Abbreviations: iPTH, intact parathyroid hormone; VMR, vitamin D metabolites ratio; 25(OH)D, 25-hydroxy vitamin D; 24,25(OH)_2_D, 24,25-dihydroxy vitamin D.

**Figure 3 nutrients-15-00578-f003:**
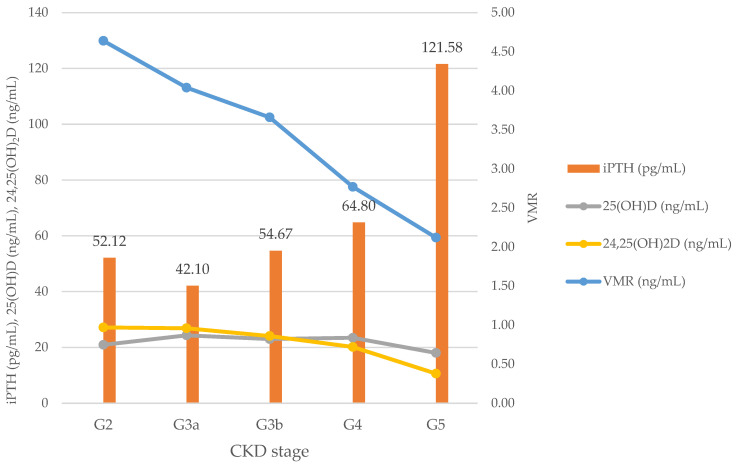
Change in vitamin D biomarkers and intact parathyroid hormone according to CKD stage. Abbreviations: CKD, Chronic Kidney Disease; iPTH, intact parathyroid hormone; VMR, vitamin D metabolites ratio; 25(OH)D, 25-hydroxy vitamin D; 24,25(OH)_2_D, 24,25-dihydroxy vitamin D.

**Table 1 nutrients-15-00578-t001:** Demographics and laboratory findings of study subjects.

	N = 208
**Age (year)**	64.3 ± 12.7
**Sex, male, (n, %)**	133 (63.9%)
**BMI (kg/m^2^)**	27.2 ± 21.5
**Comorbidities**	
DM	98 (47.1%)
HTN	167 (80.3%)
**Cause of CKD**	
Glomerular disease	
DN	45 (21.6%)
CGN	61 (33.8%)
CTIN	57 (27.4%)
Vascular disease	37 (17.8%)
PKD and other	8 (3.8%)
**Serum Cr (mg/dL)**	2.00 ± 1.07
**CKD stage (eGFR, mL/min/1.73 m^2^)**	
G2 (>60)	29 (13.9%)
G3a (45–60)	49 (23.6%)
G3b (30–44)	62 (29.8%)
G4 (15–29)	54 (26.0%)
G5 (<15)	14 (6.7%)
**Albumin (g/dL)**	4.4 ± 0.4
**Ca (mg/dL)**	9.4 ± 0.6
**P (mg/dL)**	3.8 ± 0.8
**Ca × P (mg^2^/dL^2^)**	35.1 ± 6.5
**iPTH (pg/mL)**	58.5 ± 44.4

Values are presented as mean ± SD or numbers (%). Abbreviations: BMI, body mass index; Ca, calcium; Ca × P, calcium–phosphate product; CKD, chronic kidney disease; CGN, chronic glomerulonephritis; Cr, creatinine; CTIN, chronic tubulointerstitial nephritis; DM, diabetic mellitus; DN, diabetic nephropathy; eGFR, estimated glomerular filtration rate; HTN, hypertension; iPTH, intact parathyroid hormone; P, phosphate; PKD, polycystic kidney disease.

**Table 2 nutrients-15-00578-t002:** iPTH and vitamin D biomarkers according to CKD stage.

Vitamin D Biomarkers	Total	CKD Stage					*p*
		G2	G3a	G3b	G4	G5	
iPTH (pg/mL)	58.49 ± 44.42	52.12 ± 50.00	42.10 ± 20.43	54.67 ± 29.63	64.80 ± 40.29	121.58 ± 90.14	<0.001
25(OH)D (ng/mL)	22.81 ± 11.24	20.93 ± 10.13	24.35 ± 12.37	22.99 ± 10.89	23.45 ± 11.24	18.05 ± 10.59	0.353
24,25(OH)_2_D (ng/mL)	0.83 ± 0.63	0.97 ± 0.60	0.96 ± 0.75	0.86 ± 0.58	0.72 ± 0.60	0.38 ± 0.24	0.011
VMR	3.55 ± 1.91	4.64 ± 1.98	4.04 ± 2.42	3.66 ± 1.64	2.77 ± 1.23	2.12 ± 0.85	<0.001

Abbreviations: CKD, chronic kidney disease; iPTH, intact parathyroid hormone; 25(OH)D, 25-hydroxy vitamin D; 24,25(OH)_2_D, 24,25-dihydroxy vitamin D; VMR, vitamin D metabolites ratio.

**Table 3 nutrients-15-00578-t003:** Model fitness of vitamin D biomarkers models.

Model	Explan Variables	Adjusted R^2^	AIC	BIC
Model 1: 25(OH)D = α + β_1_(eGFR) + β_2_(iPTH) + ε	eGFR, iPTH	0.022	1517.1	1530.2
Model 2: 24,25(OH)_2_D = α + β_1_(eGFR) + β_2_(iPTH) + ε	eGFR, iPTH	0.087	368.5	381.6
Model 3: VMR = α + β_1_(eGFR) + β_2_(iPTH) + ε	eGFR, iPTH	0.145	788.4	801.6
Model 4: eGFR= α + β_1_(25(OH)D) + ε	25(OH)D, iPTH	0.079	1644.2	1657.3
Model 5: eGFR = α + β_1_(24,25(OH)_2_D) + ε	24,25(OH)_2_D, iPTH	0.105	1638.5	1651.6
Model 6: eGFR = α + β_1_(VMR) + ε	VMR, iPTH	0.167	1624.4	1637.6
Model 7: iPTH= α + β_1_(25(OH)D) + ε	25(OH)D, eGFR	0.105	2037.9	2051.0
Model 8: iPTH= α + β_1_(24,25(OH)_2_D) + ε	24,25(OH)_2_D, eGFR	0.119	2034.7	2047.8
Model 9: iPTH= α + β_1_(VMR) + ε	VMR, eGFR	0.104	2038.2	2051.3

Abbreviations: eGFR, estimated glomerular filtration rate; iPTH, intact parathyroid hormone; AIC, Akaike information criterion; BIC, Bayesian information criterion; 25(OH)D, 25-hydroxy vitamin D; 24,25(OH)_2_D, 24,25-dihydroxy vitamin D; VMR, vitamin D metabolites ratio.

**Table 4 nutrients-15-00578-t004:** Vitamin D biomarkers according to the results of bone densitometry.

	Normal (n = 25)	Osteopenia (n = 14)	Osteoporosis (n = 4)	*p*
eGFR (mL/min/1.73 m^2^)	35.40 ± 19.46	33.86 ± 20.49	28.50 ± 20.86	0.552
25(OH)D (ng/mL)	23.92 ± 14.47	23.32 ± 9.41	15.88 ± 9.36	0.349
24,25(OH)2D (ng/mL)	0.84 ± 0.62	0.88 ± 0.54	0.35 ± 0.39	0.304
VMR	3.45 ± 1.60	3.55 ± 1.50	1.81 ± 1.05	0.177

Abbreviations: eGFR, estimated glomerular filtration rate; 25(OH)D, 25-hydroxy vitamin D; 24,25(OH)_2_D, 24,25-dihydroxy vitamin D; VMR, vitamin D metabolites ratio.

## Data Availability

The data utilized to support this research are available from the corresponding author upon request.

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
