# Peer review of "24,25-Dihydroxy Vitamin D and Vitamin D Metabolite Ratio as Biomarkers of Vitamin D in Chronic Kidney Disease"

_nutrients, 2023, doi:10.3390/nu15030578_

Round 1
Reviewer 1 Report
The authors present correlations of circulating 25(OH)D concentration (the substrate), 24,25(OH)2D concentration (the product), and the ratio of the product to substrate (i.e., the vitamin D metabolite ratio or VMR) with iPTH and eGFR across stages of CKD. They show that 24,25(OH)2D and VMR are more strongly correlated with eGFR and iPTH. In its present state, the manuscript fails to convince me that monitoring 24,25(OH)2D in patients with CKD or CKD-MBD has added value. The introduction does not establish what problem the authors have set out to address and does not help the reader to understand current practice for monitoring vitamin D biomarkers and related biomarkers in CKD-MBD. Nutrients is not a nephrology journal, so more information is required to justify why the research is important.
INTRODUCTION
The introduction should more clearly explain the problem that the research sought to address. You should explain what you mean by “vitamin D level” in the first paragraph. Initially, I thought you meant circulating 25(OH)D concentration. The U.S. NASEM Dietary Reference Intakes Committee (2011) described how circulating 25(OH)D concentration is a good biomarker of vitamin D exposure, but it functions less well as a biomarker of effect. However, there are additional considerations for individuals with CKD, and the introduction should be more specific to clinical monitoring in CKD.
You state, “No studies have compared the vitamin D biomarkers…in CKD”. What about reference no. 11: Bosworth et al. 2012?
MATERIALS AND METHODS
VMR. This should be the molar ratio. I recommend to convert to moles before calculating the ratio.
RESULTS
3.1. Wrong units for serum Ca and P
I think the authors should consider including 1,25(OH)2D concentrations in this manuscript. In Table 2, 24,25(OH)2D was clearly lower with stages 4 and 5 CKD. Don’t you also expect the 1,25 concentration to be lower due to loss of kidney function? Measuring 1,25 directly would tell you how much hormone is available for vitamin D endocrine function. Why should you measure 24,25 instead of 1,25 in the later stages of CKD?
DISCUSSION
How is usefulness defined? It seems to be based on associations with iPTH and eGFR, but it is routine practice to monitor those quantities in individuals with CKD and CKD-MBD. The associations are weak to moderate, so how useful is it to add another vitamin D metabolite to the laboratory order? There is also the possibility of extrarenal formation of 1,25 for autocrine/paracrine action, although the importance of that to human health has not been substantiated. If that is important, then wouldn’t maintaining adequate systemic concentrations of the substrate 25(OH)D still be important?
Author Response
Point 1: The introduction does not establish what problem the authors have set out to address and does not help the reader to understand current practice for monitoring vitamin D biomarkers and related biomarkers in CKD-MBD. Nutrients is not a nephrology journal, so more information is required to justify why the research is important.
Response 1: Yes, I agree with you about the need for an accurate explanation of monitoring vitamin D biomarkers, especially in patients with CKD.
We corrected the sentences in the third paragraph in Introduction.
“Accurate assessment of the vitamin D level is essential for the prevention and treatment of SHPT through correction of electrolyte imbalance and vitamin D supplementation. Total 25(OH)D level is commonly used to determine the vitamin D status. However, there has been debates about the reliability of this biomarker. It consists of vitamin D binding protein (VDBP)-bound 25(OH)D, albumin-bound 25(OH)D, and free 25(OH)D. They may be influenced by ethnicity, religion, sunlight exposure, and the several conditions altering vitamin D metabolism such as, obesity, drugs, and diabetes, in addition to CKD [9, 10]. Another biomarker, 1,25(OH)2D, is known to play an important role in vitamin D metabolism, functionally interacting with PTH or Fibroblast growth factor-23 (FGF-23) in CKD. However, it has a shorter half-life compared to 25(OH)D and KDIGO guidelines of CKD-MBD does not recommend to routine measure of 1,25(OH)D2D in CKD patients [11, 12].
Recently, serum 24,25(OH)2D and the ratio of 24,25(OH)2D and 25(OH)D levels (vitamin D metabolite ratio [VMR]) have been proposed as potential alternative biomarkers of vitamin D level [13, 14]. A Korean study about healthy people found that 25(OH)D is a better biomarker of vitamin D level than serum 24,25(OH)2D level and VMR [15]. However, because the production of 24,25(OH)2D depends on the 25(OH)D level and the expression of 24(OH)D hydroxylase enzyme (CYP24A1), regulated by active vitamin D, serum 24,25(OH)2D level and VMR have the potential to be alternative biomarkers of vitamin D level, particularly in CKD patients [16].” (Page 2, line 47-66)
[References]
[9] Bikle, D.; Bouillon, R.; Thadhani, R.; Schoenmakers, I. Vitamin D metabolites in captivity? Should we measure free or total 25(OH)D to assess vitamin D status? J. Steroid Biochem. Mol. Boil. 2017, 173, 105–116
[10] Castillo-Peinado, L.d.l.S.; Calderón-Santiago, M.; Herrera-Martínez, A.D.; León-Idougourram, S.; Gálvez-Moreno, M.Á.; Sánchez-Cano, R.L.; Bouillon, R.; Quesada-Gómez, J.M.; Priego-Capote, F. Measuring Vitamin D3 Metabolic Status, Comparison between Vitamin D Deficient and Sufficient Individuals. Separations 2022, 9, 141. https://doi.org/10.3390/separations9060141
Point 2: The introduction should more clearly explain the problem that the research sought to address. You should explain what you mean by “vitamin D level” in the first paragraph. Initially, I thought you meant circulating 25(OH)D concentration. The U.S. NASEM Dietary Reference Intakes Committee (2011) described how circulating 25(OH)D concentration is a good biomarker of vitamin D exposure, but it functions less well as a biomarker of effect. However, there are additional considerations for individuals with CKD, and the introduction should be more specific to clinical monitoring in CKD.
Response 2: Yes, we acknowledge that the explanation for 25(OH)D in Introduction was not sufficient. As mentioned tin the answer to question 1, we corrected several paragraphs in Introduction. Moreover, we modified the sentences in the last paragraph of Introduction, considering the reference no.11.
“There are rare studies comparing vitamin D biomarkers, including 25(OH)D, 24,25(OH)2D, and VMR in patients with CKD. In this present study, we compared various vitamin D biomarkers according to renal function in CKD patients, as well as the association of the biomarkers with intact PTH (iPTH). Since there are several limitations to the current method for evaluating vitamin D status with 25(OH)D, we aim to evaluate whether 24,25(OH)2D or VMR can be a biomarker alternative to 25(OH)D.” (Page 2, line 67-72)
Pont 3: VMR. This should be the molar ratio. I recommend to convert to moles before calculating the ratio.
Response 3: In other previous researches on VMR (ex, Nutrients. 2022 May 25;14(11):2201. doi: 10.3390/nu14112201. Clin Chim Acta. 2017 Oct;473:116-123. doi: 1016/j.cca.2017.08.024.), 25(OH)D and 24,25(OH)2D were measured in SI units such as ng/ml or ug/L by LC-MS/MS method and then VMR was calculated as in our study. Thus, we wonder whether we should calculate the VMR as a molar ratio after converting the measured values ​​of 25(OH)D and 24,25(OH)2D into moles.
Point 4: Wrong units for serum Ca and P
Response 4: We double-checked the units for the results of the Ca and P tests in our hospital laboratory and reconfirmed that they were all mg/dL
Point 5: I think the authors should consider including 1,25(OH)2D concentrations in this manuscript. In Table 2, 24,25(OH)2D was clearly lower with stages 4 and 5 CKD. Don’t you also expect the 1,25 concentration to be lower due to loss of kidney function? Measuring 1,25 directly would tell you how much hormone is available for vitamin D endocrine function. Why should you measure 24,25 instead of 1,25 in the later stages of CKD?
Response 5: As the reviewer points out, the active form of vitamin D that binds to vitamin D receptor and plays a role is 1,25(OH)2D. However, 1,25(OH)2D is not a reliable biomarker that accurately reflects vitamin D status for the following reasons. First, in fact, 1,25(OH)2D has a very short half-life of about 5 to 8 hours, thus its concentration varies greatly during the day. Second, since the blood concentration of 1,25(OH)2D is very low compared to 25(OH)D, it is difficult to accurately measure its serum concentration. For these reasons, we did not plan to measure the concentration of 1,25(OH)2D from the design of this study.
Point 6: How is usefulness defined? It seems to be based on associations with iPTH and eGFR, but it is routine practice to monitor those quantities in individuals with CKD and CKD-MBD. The associations are weak to moderate, so how useful is it to add another vitamin D metabolite to the laboratory order? There is also the possibility of extrarenal formation of 1,25 for autocrine/paracrine action, although the importance of that to human health has not been substantiated. If that is important, then wouldn’t maintaining adequate systemic concentrations of the substrate 25(OH)D still be important?
Response 6: Yes, we agree that there is a limitation to evaluating the usefulness of each vitamin D biomarker in this study. We corrected some sentences in Abstract and Conclusion.
In Abstract on Page 1, “In conclusion, 24,25(OH)2D and VMR are more useful vitamin D biomarkers than 25(OH)D for the detection of MBD in CKD patients.” was corrected to
“In conclusion, 24, 25(OH)2D and VMR have the potential to be vitamin D biomarkers for the detection of MBD in CKD patients.“ (Page 1, line 22-23)
In Conclusion on Page 9, “Therefore, we suggest that 24,25(OH)2D and VMR have the potential to be vitamin D biomarkers alternative to 25(OH)D for the diagnosis of metabolic bone disease in CKD patients.” was corrected to
“Therefore, we suggest that 24,25(OH)2D and VMR have the potential to be vitamin D biomarkers alternative to 25(OH)D for the diagnosis of metabolic bone disease in CKD patients.” (Page 9, line 289-291)
Reviewer 2 Report
Lee and colleagues present their work demonstrating that the ratio of 24,25 vitamin D (OH)2 to 25 vitamin D (OH) provides a more accurate measure of vitamin D levels than 25 vitamin D (OH) or 1,25 vitamin D (OH)2 alone in patients with progressive chronic kidney disease. They show that the ratio (vitamin D metabolite ration -VMR) - strongly correlates with the CKD stage, decreasing as CKD stage increases. They argue that this would a better way to assess metabolic bone disease in CKD. They note that the limits of the study are its cross sectional nature, and the lack of information about other confounding factors like dietary intake, sun exposure, sun screen use. They did exclude people taking vitamin d or phosphorus binders. A few comments and questions.
1. Were people also excluded if they were taking calcium supplements? Were people taking calcimimetics excluded as well?
2. It sometimes gets confusing reading about vitaminD throughout and trying to keep straight whether one is talking about 25 vit d oh or 1,25 vit d oh2. It might make it easier to read if one defines references to vit d as encompassing inactive cholecalciferol and active calcitriol and then uses those terms if referring to a specific form of vitamin d.
3. The methods are important for understanding the reproducibility of the study. But perhaps the authors include a line or two about the rational for the methods. Would this be too complicated for the average hospital lab to measure 24,25 vit d oh 2? How much cost would this add?
4. What are the practical implications of this study? Should we all run out and order 24,25 vit d oh2 and 25 vitamin d levels to calculate the VMR? What should we expect if we followed this longitudinally in individuals? Are there any plans to follow up with the participants in this study? DId any of the study participants have bone scans? Is there any sense that this ratio would correlate with bone density?
5. Is there any thought that this might be useful in the transplant population with hypercalcemia and hyperparathyroidism? Or what about post-parathyroidectomy in trying to assess how much cholecalciferol and calcitriol to give patients?
6. How reproducible would this be in other regions of the world. The authors might want to include something about average sunlight exposure in S Korea to help others decide how generalizable the findings are to their region of the world.
Author Response
Point 1: Were people also excluded if they were taking calcium supplements? Were people taking calcimimetics excluded as well?
Response 1: Yes, patients taking calcimimetics or phosphate binders, any medications and supplements containing calcium, or phosphorus as well as vitamin D, were excluded.
We corrected the sentences in 2.1 Study participants. (Page 2)
“Patients taking phosphorus binders or vitamin D supplements at the time of examination were excluded.” was corrected to
“Patients taking calcimimetics or phosphate binders, any medications and supplements containing calcium, or phosphorus as well as vitamin D, were excluded.” (Page 2, line 78-81)
Point 2: It sometimes gets confusing reading about vitamin D throughout and trying to keep straight whether one is talking about 25 vit d oh or 1,25 vit d oh2. It might make it easier to read if one defines references to vit d as encompassing inactive cholecalciferol and active calcitriol and then uses those terms if referring to a specific form of vitamin d.
Response 2: Vitamin D mentioned in the introduction or discussion of our article is a general form of vitamin D, a concept that includes all kinds of vitamin D metabolites, such as cholecalciferol, calcitriol, 25(OH)D, and 1,25(OH)2D. If it is necessary to specifically define the vitamin D metabolite in our article, it is clearly indicated it is whether 25(OH)D or 1,25(OH) 2D.
Point 3: The methods are important for understanding the reproducibility of the study. But perhaps the authors include a line or two about the rational for the methods. Would this be too complicated for the average hospital lab to measure 24,25 vit d oh 2? How much cost would this add?
Response 3: To aid readers' understanding, one or two sentences of explanations on the methods were added to Materials and Methods. In addition, special equipment and skilled operators are required to measure 24,45 (OH)2D or 25(OH)D by LC-MS/MS method. And 24,45 (OH)2D is measured only for research purposes. Our study provided the possibility that VMR or 24,25(OH)D could be used as a useful vitamin D marker in CKD patients. It seems that more large-scale research is needed to use it for patient care. Thus, it seems premature to discuss the price of the test when applied to patients.
Point 4: What are the practical implications of this study? Should we all run out and order 24,25 vit d oh2 and 25 vitamin d levels to calculate the VMR? What should we expect if we followed this longitudinally in individuals? Are there any plans to follow up with the participants in this study? DId any of the study participants have bone scans? Is there any sense that this ratio would correlate with bone density?
Response 4: The purpose of this study is to suggest the potential as a biomarker of 24,25(OH)2D and VMR, and further studies are still needed for clinical application. A follow-up study including bone densitometry seems necessary. 43 patients performed bone densitometry with the consent. In patients with osteoporosis, vitamin D biomarkers tended to decrease. However, the number of patients was too small, and vitamin D biomarkers did not show significant differences according to bone density.
We will provide their results as a supplementary data. (Page 10) We added some sentences about the results of bone densitometry in Discussion.
“In this presented study, 43 patients performed bone densitometry and the results is provided in Table S2. In osteoporosis patients, vitamin D biomarkers tended to decrease. However, the number of patients was too small, and vitamin D biomarkers did not show significant differences according to bone density. A follow-up study including more data about bone densitometry seems necessary to evaluate the correlation between bone density and vitamin D biomarkers. “ (Page 9, line 259-264)
|
Table S2. Vitamin D biomarkers according to the results of bone densitometry.   |
||||
|
  |
Normal (n=25) |
Osteopenia (n=14) |
Osteoporosis (n=4) |
p |
|
eGFR (mL/min/1.73m2) |
35.40±19.46 |
33.86±20.49 |
28.50±20.86 |
0.552 |
|
25(OH)D (ng/mL) |
23.92±14.47 |
23.32±9.41 |
15.88±9.36 |
0.349 |
|
24,25(OH)2D (ng/mL) |
0.84±0.62 |
0.88±0.54 |
0.35±0.39 |
0.304 |
|
VMR |
3.45±1.60 |
3.55±1.50 |
1.81±1.05 |
0.177 |
|
Abbreviations: eGFR, estimated glomerular filtration rate; 25(OH)D, 25-hydroxy vitamin D; 24,25(OH)2D, 24,25-dihydroxy vitamin D; VMR, vitamin D metabolites ratio. |
||||
Point 5: Is there any thought that this might be useful in the transplant population with hypercalcemia and hyperparathyroidism? Or what about post-parathyroidectomy in trying to assess how much cholecalciferol and calcitriol to give patients?
Response 5: Thanks for the good suggestions. Our study mainly aimed to compare and evaluate the vitamin D biomarkers that best reflects vitamin D status in CKD patients without dialysis or transplantation. A study on patients with hypercalcemia and hyperparathyroidism after kidney transplantation or post-parathyroidectomy as mentioned by the reviewer would be very interesting research topics. However, these topics are beyond the scope of our research, thus it does not seem appropriate to mention them in this paper. If our research team may enroll a sufficient number of patients for the study, we will consider those topics as a follow-up study.
Point 6: How reproducible would this be in other regions of the world. The authors might want to include something about average sunlight exposure in S Korea to help others decide how generalizable the findings are to their region of the world.
Response 6: Unlike other vitamin D biomarkers, VMR does not show significant differences between races, and is known to be a better biomarker than 25(OH)D, especially in cases of vitamin D deficiency (PMID: 28842174). Therefore, it seems possible to apply the VMR to other ethnic group. In addition, according to the website (http://www.seoul.climatemps.com/sunlight.php), which provides year-round weather statistics for Seoul, the capital of South Korea, there is an average of 2428 hours of sunlight per year (of a possible 4383) with an average of 6:38 of sunlight per day. However, besides geographical factor such as average sunlight exposure, the serum vitamin D concentration is influenced by other various factors such as genetic predisposition, age, use of sun screen, and dietary intake. Therefore, in order to globalize our research results, further sophisticated and large-scaled researched might be required.
We added the above sentences to the Discussion. (Page 9, line 291-300)
“VMR does not show significant differences between races, and is known to be a better biomarker than 25(OH)D, especially in cases of vitamin D deficiency [32]. Therefore, it is possible to apply the VMR to other ethnic group. In addition, according to the website, which provides year-round weather statistics for Seoul, the capital of South Korea, there is an average of 2428 hours of sunlight per year (of a possible 4383) with an average of 6:38 of sunlight per day [33]. However, besides geographical factor such as average sunlight exposure, the serum vitamin D concentration is influenced by other various factors such as genetic predisposition, age, use of sun screen and dietary intake. Therefore, in order to globalize our research results, further sophisticated and large-scaled research might be required”
[Reference]
[32] Fabregat-Cabello N, Farre-Segura J, Huyghebaert L, et al. A fast and simple method for simultaneous measurements of 25(OH)D, 24,25(OH)2D and the Vitamin D Metabolite Ratio (VMR) in serum samples by LC-MS/MS. Clin Chim Acta. 2017;473:116-123. doi:10.1016/j.cca.2017.08.024
[33] http://www.seoul.climatemps.com/sunlight.php
Round 2
Reviewer 1 Report
The authors have made important changes to the manuscript, but the clinical question and relevance of the findings are still obscured. I think the introduction and discussion can be revised to increase clarity. The authors are looking for an improved biomarker of "vitamin D level". What do they mean by that? What is the physiological phenomenon that they want to know about? There are several possible definitions of "vitamin D level": (1) Substrate supply, i.e., the pre-prohormone vitamin D and (more importantly) prohormone 25(OH)D available; (2) Hormone 1,25 available systemically/intracellularly for hormone action; (3) The degree to which physiological needs for 1,25 have been satisfied. There may be other possibilities. Again, what exactly do these authors want to know about their CKD patients, and does that change as CKD progresses? I read the KDIGO CKD-MBD guidelines from 2009 and the update from 2017. The guidelines recommend monitoring circulating P, Ca, PTH, and ALP. Circulating 25(OH)D might be monitored to screen for nutritional vitamin D deficiency (substrate deficiency) because that could be exacerbating the MBD and can be corrected. KDIGO guidelines suggest that vitamin D deficiency and insufficiency defined according to circulating 25(OH)D be corrected using treatment strategies recommended for the general population. In advanced CKD, 25(OH)D supply becomes uncoupled from PTH and functional vitamin D status. In that situation, what is it that the authors wish to know about their patients?
Author Response
Point 1: The authors have made important changes to the manuscript, but the clinical question and relevance of the findings are still obscured. I think the introduction and discussion can be revised to increase clarity. The authors are looking for an improved biomarker of "vitamin D level". What do they mean by that? What is the physiological phenomenon that they want to know about? There are several possible definitions of "vitamin D level": (1) Substrate supply, i.e., the pre-prohormone vitamin D and (more importantly) prohormone 25(OH)D available; (2) Hormone 1,25 available systemically/intracellularly for hormone action; (3) The degree to which physiological needs for 1,25 have been satisfied. There may be other possibilities. Again, what exactly do these authors want to know about their CKD patients, and does that change as CKD progresses? I read the KDIGO CKD-MBD guidelines from 2009 and the update from 2017. The guidelines recommend monitoring circulating P, Ca, PTH, and ALP. Circulating 25(OH)D might be monitored to screen for nutritional vitamin D deficiency (substrate deficiency) because that could be exacerbating the MBD and can be corrected. KDIGO guidelines suggest that vitamin D deficiency and insufficiency defined according to circulating 25(OH)D be corrected using treatment strategies recommended for the general population. In advanced CKD, 25(OH)D supply becomes uncoupled from PTH and functional vitamin D status. In that situation, what is it that the authors wish to know about their patients?
Response 1: Thank you again for your suggestion.
In order to change our views more clearly and in detail, some sentences of Introduction and Discussion were modified and added.
In introduction,
“In addition, previous observational studies and randomized trials suggested that serum 25(OH)D may not be an accurate biomarker of vitamin D status, especially in bone health [11-13]. In fact, the 25-hydroxylation reaction in the liver does not involve a delicate feedback process to control the vitamin D level in the body, whereas the process of 1α-hydroxylation synthesizing 1,25(OH)2D in the kidney involves many feedback processes via vitamin D receptor (VDR) [4]. In this respect, the biomarker that most accurately reflects vitamin D status in the body would be 1,25(OH)2D. 1,25(OH)2D is known to play an important role in vitamin D metabolism and functionally interacts with PTH or fibroblast growth factor-23 (FGF-23) in CKD. However, it has a very short half-life of about 5 to 8 hours, thus its concentration varies greatly during the day. In addition, since the blood concentration of 1,25(OH)2D is relatively low, it is difficult to accurately measure its serum concentration [14, 15].” (Page 2, line 56-67)
“24,25(OH)2D is a catabolic product by a negative feedback mechanism to control the concentration of vitamin D in the blood. There is a complicated feedback mechanism to control blood vitamin D level. In case of vitamin D toxicity, negative feedback turns on the catabolism mechanism to reduce blood vitamin D levels. The activated 24(OH)D hydroxylase enzyme (CYP24A1) converted 25(OH)D to 24,25(OH)2D by the stimulation of 1,25(OH)2D [20, 21]. Therefore, 24,25(OH)2D or VMR could be a physiologically better biomarker than the currently used 25(OH)D in that they represent the result of a feedback mechanism. In addition, previous studies have suggested that VMR may have stronger association with bone health than that of 25(OH)D [22, 23]. Kidney is a major organ in which CYP24A1 are expressed and activated. When evaluating vitamin D status in CKD patients, it may be necessary to consider not only 25(OH)D level but also that the activity of CYP24A1 may change depending on kidney function. For this reason, 24,25(OH)2D level or VMR might be biomarkers of vitamin D status, particularly in CKD patients [24].” (Page 2, line 71-84)
We revised the sentences in Discussion, considering your suggestions in the previous review. Also, we added a sentence to emphasize the results of Figure 3.
“In the present study, we compared biomarkers of the vitamin D level in patients with CKD, particularly to determine the usefulness of 24,25(OH)2D and VMR as the alternative biomarkers of 25(OH)D.” (Page 9, line 246-248)
“Our study suggests that 24,25(OH)2D and VMR may be more sensitive vitamin D biomarkers than 25(OH)D in CKD patients because they started to decrease earlier stage than 25(OH)D in Figure 3.” (Page 10, line 290-292)
Reviewer 2 Report
Thank you for answering my queries, particularly about bone density an reproducibility of study
Author Response
Point 1: Thank you for answering my queries, particularly about bone density an reproducibility of study
Response 1: Thank you again for your suggestions that we could add important information about bone density and reproducibility.
Round 3
Reviewer 1 Report
The authors addressed my comments. No further recommendations.